# Dominant drug targets suppress the emergence of antiviral resistance

Elizabeth J Tanner[1], Hong-mei Liu[2], M Steven Oberste[2], Mark Pallansch[2], Marc S Collett[3], Karla Kirkegaard[1]*

[1]Department of Microbiology and Immunology, Stanford University School of Medicine, Stanford, United States; [2]Division of Viral Diseases, Centers for Disease Control and Prevention, Atlanta, United States; [3]ViroDefense, Inc., Rockville, United States

**Abstract** The emergence of drug resistance can defeat the successful treatment of pathogens that display high mutation rates, as exemplified by RNA viruses. Here we detail a new paradigm in which a single compound directed against a 'dominant drug target' suppresses the emergence of naturally occurring drug-resistant variants in mice and cultured cells. All new drug-resistant viruses arise during intracellular replication and initially express their phenotypes in the presence of drug-susceptible genomes. For the targets of most anti-viral compounds, the presence of these drug-susceptible viral genomes does not prevent the selection of drug resistance. Here we show that, for an inhibitor of the function of oligomeric capsid proteins of poliovirus, the expression of drug-susceptible genomes causes chimeric oligomers to form, thus rendering the drug-susceptible genomes dominant. The use of dominant drug targets should suppress drug resistance whenever multiple genomes arise in the same cell and express products in a common milieu.

*For correspondence: karlak@stanford.edu

## Introduction

Drug-resistant variants of positive-strand RNA viruses are generated due to error-prone replication of parental, drug-susceptible genomes. Even if individual host cells are infected with only a single virus each, error rates of $1 \times 10^{-4}$ per nucleotide or higher ensure that each cell will experience a mixed infection as variant progeny genomes arise within it (*Holland et al., 1989*). The new variants act as templates for both continued genome synthesis and translation of new proteins. What happens in the face of a new selective pressure, such as the addition of an antiviral drug? In most cases, antiviral treatments prevent the growth of drug-susceptible viral genomes, but any drug-resistant variants in the same cell continue to amplify according to their fitness (*Figure 1A*, top).

In contrast, the dominant drug target approach presented here exploits the potential interactions between the products of drug-resistant and drug-susceptible genomes within the same cell that render the drug-susceptible phenotype dominant (*Crowder and Kirkegaard, 2005*). The most intuitive scenario for such dominance involves highly oligomeric assemblages such as viral capsids, because drug-resistant subunits will assemble with drug-susceptible ones, and the chimeric structure is likely to be inhibited by the drug (*Figure 1A*, bottom). This situation is akin to a dominant–negative interaction and examples of such dominance by vector-mediated expression of mutant proteins are abundant. Expression of a non-functional version of the oligomeric Tat protein of HIV drastically reduces viral yield (*Meredith et al., 2009*). Similarly, co-expression of non-functional NS5A protein of hepatitis C virus inhibits replication of genomes that encode functional NS5A (*Graziani and Paonessa, 2004*). In poliovirus, viral genomes with mutations in several proteins, including capsid proteins and the RNA-dependent RNA polymerase, interfere with the amplification of co-transfected wild-type viral genomes (*Crowder and Kirkegaard, 2005*).

**eLife digest** Treating a viral infection with a drug sometimes has an unwanted side effect—the virus quickly becomes resistant to the drug. Viruses whose genetic information is encoded in molecules of RNA mutate faster than DNA viruses and are particularly good at developing resistance to drugs. This is because the process of copying the RNA is prone to errors, and by chance some of these errors, or mutations, may allow the virus to resist the drug's effects.

Treating viral infections with most drugs destroys the viruses that are susceptible to the drug and inadvertently 'selects' for viruses that are resistant to the drug's effects. These drug-resistant viruses are harder to treat and often require physicians to switch between different drugs. Sometimes these new drug-resistant viruses spread and these new infections cannot be treated with drugs that would have worked in the past. So far, the best strategy to prevent drug-resistant viruses from growing in patients is to use multiple drugs, such as the life-saving treatments for HIV infection. However, for many viral infections—such as those that cause the common cold, dengue fever, Ebola, and polio—no drugs are yet available to treat infected people. Moreover, there are concerns that, if a new drug is used on its own, the viruses will quickly develop resistance to the drug and render it ineffective.

Tanner et al. now show that an antiviral drug that interferes with the formation of the outer layer (or capsid) of the poliovirus inhibits the emergence of drug resistance. The drug, called V-073, is currently being tested as a treatment for poliovirus and will be useful in the worldwide eradication effort. Tanner et al. show that treating poliovirus-infected mice with V-073 does not select for drug-resistant strains of the virus—and provide evidence that this occurs because the drug targets an assemblage of proteins.

The poliovirus capsid is assembled from a mix of proteins from different naturally occurring strains of the virus within the infected cell. A new strain of virus is always 'born' into a cell that is already infected by other viruses, which could be thought of as its parents, cousins and siblings. A new drug-resistant virus will therefore be forced to mix its capsid proteins with those of its 'family' members, who are all drug-sensitive. These hybrid capsids will remain vulnerable to the drug—and in this way, the resistant strains do not become the dominant form of the virus.

Tanner et al. also discovered a way to screen for drugs that have a similar resistance-blocking effect. These drugs would target capsids, or other viral structures made up of a mix of proteins from different virus strains. Such drugs might be useful against other viruses including the ones that cause the common cold, hepatitis C, or dengue fever.

The idea of dominant drug targeting differs from conventional dominant–negative interaction experiments in two ways. First, the dominant products are encoded by the wild-type, drug-susceptible genome that initiated the infection as well as its drug-susceptible progeny. The second unique feature is that the mixed infection of drug-susceptible and drug-resistant viruses does not result from co-infection or co-transfection. Instead, the intracellular diversity arises from the error-prone replication of the viruses themselves, even if the infection in any individual cell is initiated by a single viral genome. Here, we show that V-073, an inhibitor of viral capsid function that is currently in development for use in the poliovirus eradication campaign (*Buontempo et al., 1997*; *Collett et al., 2008*; *Oberste et al., 2009*), fulfills the expectations of dominant drug targeting, dramatically suppressing the growth of drug resistant viruses in murine infections and in cell culture via the formation of mixed assemblages.

## Results

### Growth of wild-type virus during treatment with guanidine allows selection of guanidine-resistant viruses in mice

The active sites of enzymes are often considered to be promising drug targets, due to a wealth of biochemical analyses and thousands of precedents. For poliovirus, one such target is 2C protein, a membrane-associated NTPase that is required for viral genome replication. The NTPase activity of protein 2C is inhibited by low concentrations of guanidine in cell culture and in solution (*Pfister and Wimmer, 1999*). We investigated the effectiveness of guanidine in inhibiting viral growth in mice and

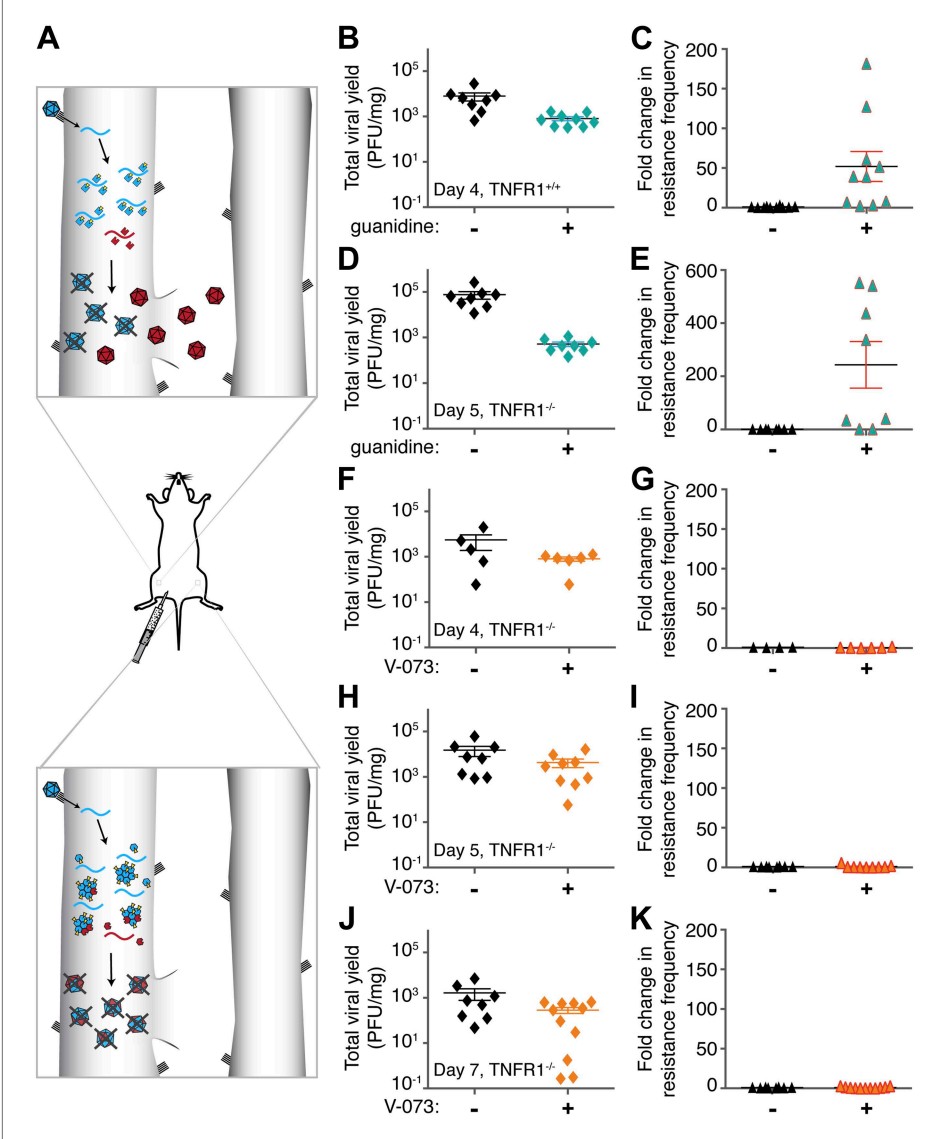

**Figure 1**. Emergence of drug-resistant variants in non-dominant and dominant drug targets in mice. (**A**) Representation of drug-resistant viral amplification when a non-dominant drug target is inhibited (top) but its inhibition by drug-susceptible variants when a dominant drug target, such as an oligomeric structure, is inhibited (bottom). See text for discussion. (**B–K**) *Tnfr1⁺/⁺* (**B**, **C**) or *Tnfr1⁻/⁻* (d-k) PVR-expressing mice were infected with 1 × 10⁷ PFU Mahoney type 1 poliovirus by intramuscular inoculation and treated with 76 mg/kg/day guanidine (**B–E**, green symbols) or 10 mg/kg/day V-073 (1-[(2-chloro-4-methoxyphenoxy)methyl]-4-[(2,6-dichlorophenoxy)methyl]benzene) (**F–K**, orange symbols). Inoculated muscles were harvested at times indicated. Total viral yields (PFU/mg tissue) in muscle samples of mice treated with guanidine (**B**, **D**), V-073 (**F**, **H**, **J**) or vehicles (black) are shown. Fold changes in the frequency of drug-resistant variants in mice treated with guanidine (**C**, **E**) or V-073 (**G**, **I**, **K**) are shown. Fold change calculated by dividing sample value by mean value of control mice. p-values: 0.02 (**B**), 0.003 (**C**), 0.0002 (**D**) 0.02 (**E**), and 0.01 (**F**, **H**, **J** combined).

whether treatment with guanidine caused the selection of compound-resistant viruses in a mouse model of poliovirus infection (*Ren and Racaniello, 1992*; *Crotty et al., 2002*). Transgenic mice that express the human poliovirus receptor (*Mendelsohn et al., 1989*) under the control of the murine actin promoter (*Crotty et al., 2002*) were bred to be homozygously deleted for TNF receptor 1 in order to delay poliomyelitis in infected mice (See results below). Subcutaneous delivery of guanidine began immediately after infection and muscle tissue was harvested at 4 or 5 days post infection.

Virus stocks were made from tissue and titered. Guanidine treatment caused significant reductions in total viral yield in $Tnfr1^{+/+}$ $PVR^{+/+}$ mice after 4 days of viral infection (*Figure 1B*) and in $Tnfr1^{-/-}$ $PVR^{+/+}$ mice after 5 days of infection (*Figure 1D*). These viral stocks were then titered in the presence of guanidine. Due to the heterogeneous nature of the viral population, even samples from untreated mice contain a measureable background frequency of guanidine-resistant variants. These variants naturally occur at a frequency of approximately two in one thousand. The frequency of guanidine resistance in each experimental sample is presented as the fold change from this average value. Large and significant increases in the frequency of guanidine resistance were observed in all treated mice (*Figure 1C,E*). Therefore, newly arising guanidine-resistant virus could be readily selected during guanidine treatment in mice infected with poliovirus. This selection for drug resistance mimics the treatment-dependent emergence of drug resistance observed in patients infected with rapidly evolving viruses. Targeting the active sites of monomeric enzymes is likely to select for drug resistance.

## Growth of wild-type virus during treatment with V-073 does not cause outgrowth of V-073-resistant viruses in mice

The dominant drug target hypothesis predicts that the poliovirus capsid, a highly oligomeric structure, is very likely to be a dominant drug target (*Crowder and Kirkegaard, 2005*). For several picornaviruses, the function of viral capsid can be inhibited by 'WIN' compounds, which have sixty binding sites on each virion (*Diana et al., 1985*; *Smith et al., 1986*), These binding sites are hydrophobic pockets arrayed about each of the twelve fivefold axes (*Diana et al., 1985*; *Smith et al., 1986*). Drug binding reduces capsid flexibility thereby preventing the conformational changes required for cell entry and capsid uncoating (*Lewis et al., 1998*; *Dove and Racaniello, 2000*). A chemically related drug, V-073, inhibits all three poliovirus serotypes and is currently in clinical development (*Buontempo et al., 1997*; *Collett et al., 2008*; *Oberste et al., 2009*).

To test the ability of V-073 to inhibit poliovirus growth and to select for resistant virus in mice, infected $Tnfr1^{-/-}$ $PVR^{+/+}$ mice were treated with V-073 by oral gavage as described previously (*Buontempo et al., 1997*). As with guanidine-treated mice, the drug regimen began immediately following infection and muscle samples were collected and titered in the absence and presence of the drug. V-073 treatment caused a significant decrease in viral titer after 4, 5 and 7 days of infection (*Figure 1F,H, J*). However, no increase in the frequency of V-073 resistance was observed at any time point (*Figure 1G,I, K*). Therefore, the selection of V-073 resistance, unlike selection for guanidine-resistant variants, did not occur during murine infection. V-073 capsid inhibition provides the first in vivo example of a potential dominant drug target interaction.

## Fitness defects do not account for the lack of V-073 resistance in mice

It was possible that a very high fitness cost accounted for the failure of V-073-resistant variants to emerge during drug treatment. To test the fitness of guanidine- and V-073-resistant viruses directly, we isolated such variants from the infected muscle tissue of mice inoculated with wild-type virus. The presence of drug-resistant viruses in the absence of any selective pressure results from the natural variation in the viral pools. Both guanidine- and V-073-resistant variants naturally occurred at a frequency of approximately one in ten thousand in samples from untreated mice. Pooled variants of both types were selected and amplified in cell culture. Drug-resistant variants grown in cell culture were propagated at low infectious doses, so that each cell was infected with no more than one virus. When these stocks were used to infect mice (*Figure 2A*), the guanidine-resistant viruses were viable but displayed significantly reduced fitness compared to wild-type virus. On the other hand, only a slight and statistically insignificant difference in fitness between wild-type virus and the V-073-resistant virus stock was observed. Thus, guanidine-resistant viruses emerged readily under selection pressure even though their fitness is low. V-073-resistant variants did not emerge in spite of showing very little growth defect. Therefore, we conclude that fitness costs do not account for the lack of outgrowth of V-073-resistant virus. Furthermore, the fact that guanidine-resistant variants are readily selected during treatment despite their fitness cost suggest that, at least in this example, relying on high fitness cost to prevent the emergence of resistance is insufficient; additional factors must be at play to prevent the emergence of resistance from a single drug regimen.

To test another example of potential selection pressure for drug-resistant viruses, we monitored the viral populations present at the time that paralytic symptoms developed. Mice that express the human poliovirus receptor (*Crotty et al., 2002*) and retain normal TNF signaling developed paralysis

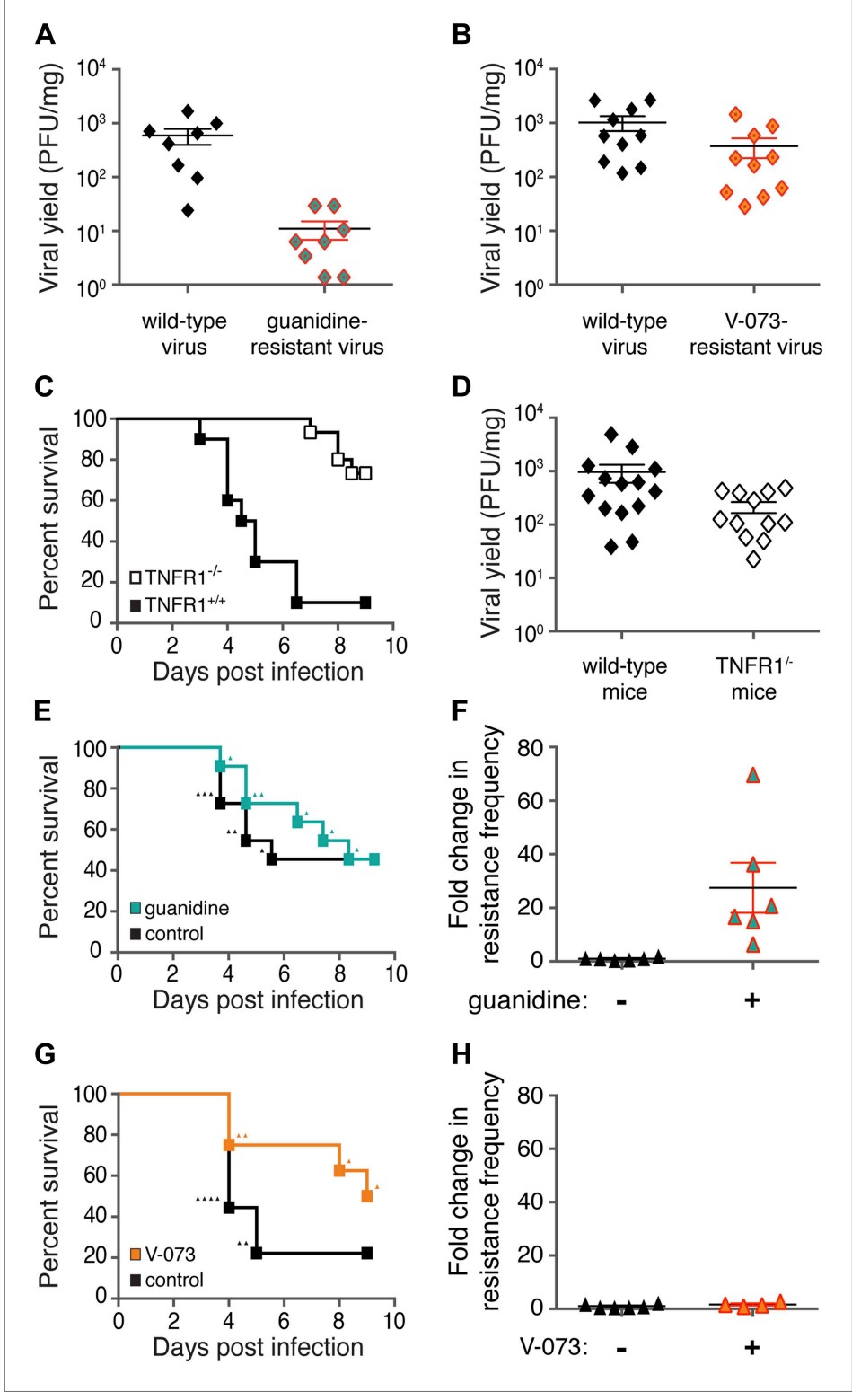

**Figure 2**. Differences in frequencies of guanidine-resistant and V-073-resistant viruses do not result from differences in viral fitness or murine health. (**A**, **B**) Total viral yield (PFU/mg) of mouse-adapted wild-type and guanidine-resistant virus (p = 0.0006) or wild-type and V-073-resistant virus in muscles of *Tnfr1*^−/− *PVR*^+/+^ mice 4 days post infection. (**C**) Survival of *Tnfr1*^+/+^ and *Tnfr1*^−/−^ mice after infection. *Tnfr1*^+/+^, n = 10; *Tnfr1*^−/−^ mice, n = 15. p = <0.0001. (**D**) Viral yield (PFU/mg) in muscles of *Tnfr1*^−/−^ mice compared to *Tnfr1*^+/+^ mice 4 days post infection. p = 0.035. *Figure 2. Continued on next page*

*Figure 2. Continued*

(**E**, **F**) Survival (**E**) and frequency of drug resistance (**F**) of cPVR mice in the presence and absence of treatment with 76 mg/kg/day guanidine. For both groups in survival analysis, n = 11. Triangles in (**E**) indicate individual mice that are analyzed in (**F**). (**G**, **H**) Survival (**G**) and frequency of drug resistance (**H**) of cPVR mice in the presence and absence of treatment with 3 mg/kg/day V-073. For survival analysis, untreated mice, n = 9; treated mice n = 8. Triangles in (**G**) indicate individual mice that are analyzed in (**H**). Mouse tissue was harvested immediately upon the first observation of paralysis (**E**–**H**). All mice were inoculated with $1 \times 10^6$ PFU virus.

beginning at 3 days post-infection (*Figure 2C*). This is in contrast to $Tnfr1^{-/-}$ $PVR^{+/+}$ mice, which displayed reduced pathogenesis (*Figure 2C*). To test whether drug resistance had broken through in mice that developed poliomyelitis, muscle tissue from infected mice was harvested as soon as the animal developed signs of paralysis. When testing drug resistance in $Tnfr1^{+/+}$ mice, the yield of total virus after 4 days of infection was higher than in $Tnfr1^{-/-}$ mice (*Figure 2D*), perhaps providing more viral diversity on which selection pressure could act during drug treatment.

Treatment of poliovirus-infected transgenic mice with guanidine throughout a course of infection did not affect pathogenesis (*Figure 2E*). However, the compound certainly had some effect: in mice that developed poliomyelitis, large and significant increases in the frequency of guanidine-resistant virus were observed (*Figure 2F*). On the other hand, when the few mice that developed paralysis during V-073-treatment were analyzed, no selection for V-073-resistant virus was seen (*Figure 2G*). We conclude that, during viral growth under either extended periods of selection or conditions that led to disease, viruses resistant to guanidine were readily selected in mice while viruses resistant to V-073 were not. These data argue that treatment-dependent emergence of resistance is not an inevitable outcome of antiviral monotherapy. In this example, an increase in resistance was never observed despite the fact that drug-resistant variants with relatively high fitness arose throughout the course of infection. We propose that the poliovirus capsid constitutes the first confirmed case of a dominant drug target.

## The viral capsid, but not 2C NTPase or 3C proteinase, is a dominant drug target during co-infection in cell culture

To test directly whether drug-susceptible viruses were dominant over drug-resistant viruses in the cases of guanidine, V-073 and a proteinase inhibitor, rupintrivir, we used deliberate co-infection in cell culture. If drug-susceptible variants dominantly interfere with the growth of drug-resistant variants, the yield of the resistant variant should decrease when co-infected with drug-susceptible virus. For guanidine (*Figure 3A*), a known variant, 2C-N179G (*Pincus and Wimmer, 1986*), was reconstructed and shown to confer complete resistance to 0.5 mM guanidine (*Figure 3B*). To mimic the mix of genomes in which a new drug-resistant variant would first arise, HeLa cells were co-infected with 2C-N179G virus and increasing amounts of wild-type, drug-susceptible virus in the presence of guanidine. The multiplicity of infection (MOI) of the drug-resistant 2C-N179G variant was maintained at 10 plaque-forming units (PFU)/cell while that of the drug-susceptible virus ranged from 0 to 100 PFU/cell. As shown in *Figure 3C*, the yield of guanidine-resistant virus in the presence of the drug was the same whether susceptible viruses were present or not. Even when the guanidine-resistant variant was outnumbered 100:1, its growth was unimpeded by the presence of wild-type, drug-susceptible virus (*Figure 3D*). Therefore, targeting the viral NTPase with guanidine allows the growth of drug-resistant genomes even in the presence of an excess of drug-susceptible ones. This is consistent with the results in mice, in which guanidine-resistant variants were readily selected.

To test whether viruses susceptible to V-073 (*Figure 3E*) were genetically dominant in cell culture, we selected a V-073-resistant variant of Mahoney type 1 poliovirus by repeated passage at low multiplicities of infection so that any pre-existing genome that was V-073-resistant could be amplified. Such variants were readily isolated and the responsible mutation, VP3-A24V (*Figure 3F*) was identified. Unlike with guanidine, co-infection of wild-type, drug-susceptible virus with V-073 resistant virus resulted in a dominant inhibition in drug-resistant virus growth (*Figure 3G*). As in mice, this did not result from the decreased fitness of V-073-resistant virus (*Figure 3H*). Even at only a fivefold excess of V-073-susceptible virus, the yield of drug-resistant virus was reduced by 98%. A new drug-resistant variant created within a cell is likely to be greeted by an excess of drug-susceptible genomes much greater than this 5:1 ratio, making genetic dominance of the drug-susceptible virus a viable mechanism for the suppression of V-073 resistance in the mouse model.

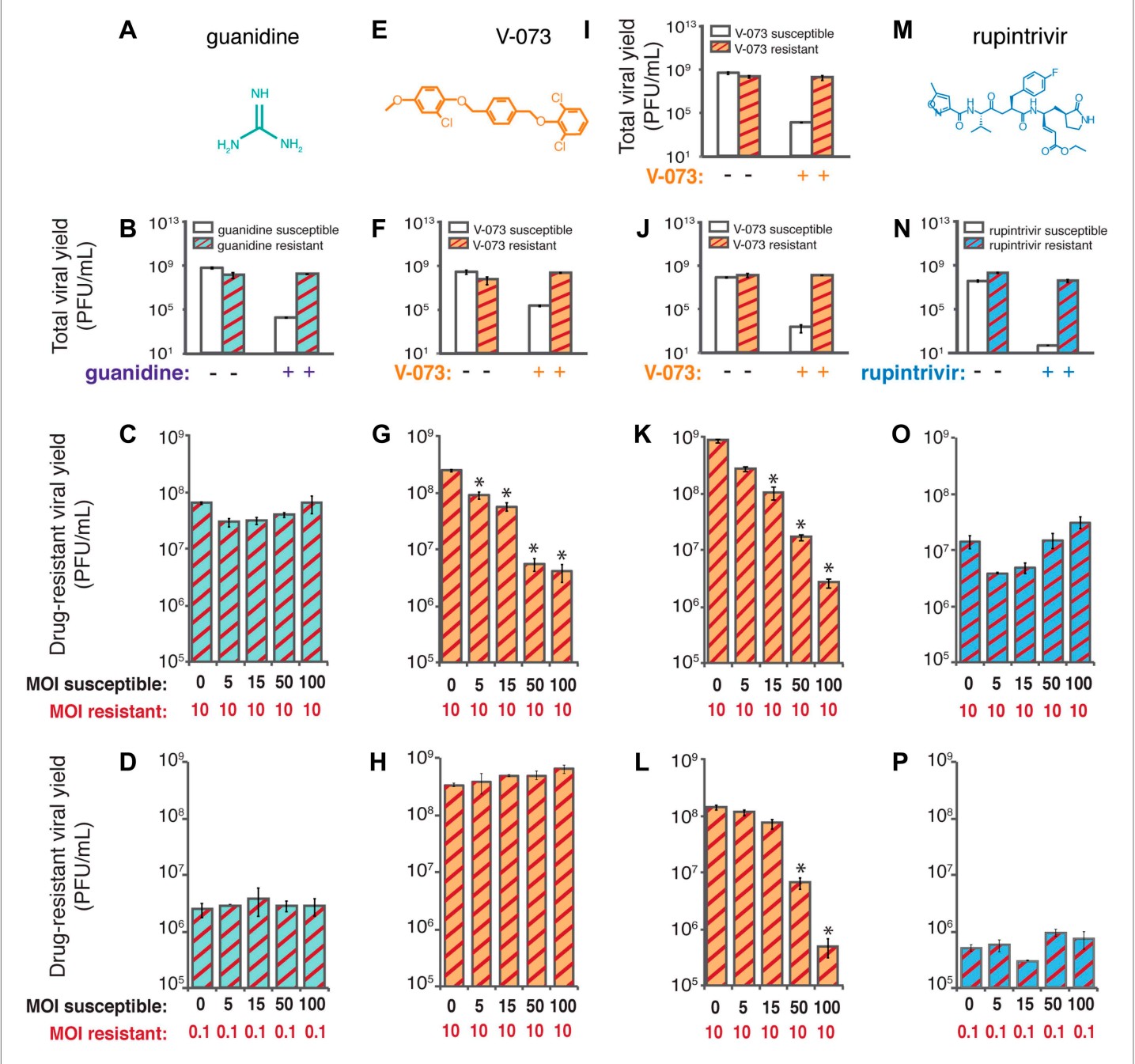

**Figure 3**. Co-infection experiments to test genetic dominance of drug-susceptible genomes. Differential growth of defined viral variants resistant to compounds inhibitory to three different viral targets in mixed infections. (**A**) Chemical structure of guanidine, which inhibits the NTPase activity of poliovirus 2C protein. (**B**) A previously characterized guanidine-resistant poliovirus variant (2C-N179G) was genetically reconstructed. The yield of 2C-N179G virus at (**C**) MOI of 10 PFU/cell and (**D**) MOI of 0.1 PFU/cell in the presence of guanidine and increasing amounts of wild-type, drug-susceptible virus in the same cells was determined. (**E**) Chemical structure of V-073, which inhibits the function of the icosahedral poliovirus capsid (**F**) A V-073-resistant poliovirus variant (VP3-A24V) was identified and genetically reconstructed in the Mahoney type 1 viral background. The yield of VP3-A24V virus in the (**G**) presence and (**H**) absence of V-073 and increasing amounts of wild-type, drug-susceptible virus in the same cells was determined. (**I**) V-073-resistant variant VP3-A24V and (**J**) VP1-I94F in the epidemic Hispaniola strain of poliovirus were identified and reconstructed. The yields of (**K**) VP3-A24V and (**L**) VP1-I94F Hispaniola type 1 poliovirus in the presence of V-073 and increasing amounts of wild-type, drug-susceptible Hispaniola virus in the same cells were determined. (**M**) Chemical structure of rupintrivir (trans-(4S,2R,5S,3S)-4-{2-4-(4-fluorobenzyl)-6-methyl-5′-[(5-methylisoxazole)-3-carbonylamino]-4-oxoheptanoylamino}-5-(2-oxopyrrolidin-3-yl)pent-2-enoic acid ethyl ester), which targets the active site of poliovirus 3C proteinase. (**N**) A rupintrivir-resistant poliovirus variant (3C-G128Q) was identified and genetically reconstructed. The yield of 3C-G128Q virus at (**O**) an MOI of 10 PFU/cell and at (**P**) 0.1 PFU/cell in the presence of rupintrivir and increasing amounts of wild-type, drug-susceptible virus in the same cells was determined. *p < 0.01.

To determine whether selection for V-073-resistant virus was also suppressed in a viral strain isolated from a poliomyelitis patient (*Kew et al., 2002*), we tested whether the Hispaniola strain of vaccine-derived neurovirulent poliovirus, known to be susceptible to V-073 (*Oberste et al., 2009*), was dominant over two different V-073-resistant variants. One such variant harbored the VP3-A24V mutation mentioned above and the second contained an I194F mutation in another capsid protein, VP1 (*Kouiavskaia et al., 2011*; *Liu et al., 2012*). Each of these mutations alters residues that line the pocket known to bind V-073, confers drug resistance and allows growth comparable to wild-type virus in the absence of drug (*Figure 3I,J*). Like the laboratory-passaged Mahoney type 1 strain (*Figure 3G*), these V-073-resistant Hispaniola-derived variants were profoundly inhibited by the presence of the parental strain (*Figure 3K,L*). Therefore, drug-susceptible virus was genetically dominant over drug-resistant variants in both lab-passaged and disease-causing strains, suggesting that V-073 is a general dominant drug target inhibitor.

Viral proteinases are often effective antiviral drug targets (*Malcolm, 1995*; *Patick and Potts, 1998*), so it was of interest to determine whether inhibition of the major proteinase of poliovirus, termed 3C, was a dominant drug target. Rupintrivir (*Figure 3M*) is an active site inhibitor of enterovirus 3C proteinases (*Patick et al., 1999*, *2005*; *Wang et al., 2011*), and is, in cell culture, the most effective inhibitor of poliovirus growth tested here (*Figure 3N*). Although no rupintrivir-resistant viruses have been reported previously, we could select such a variant during multiple viral passages in the presence of the drug. The phenotype of the rupintrivir-resistant variant (*Figure 3N*) was caused by a single amino acid change, G128Q, in the 3C coding region. As was the case with guanidine, the yield of rupintrivir-resistant virus grown in the presence of rupintrivir was relatively unperturbed by the presence of susceptible virus (*Figure 3O*), up to ratios of 100:1 of susceptible to resistant virus (*Figure 3P*). Thus, viral proteinase 3C, like viral NTPase 2C, is a non-dominant drug target.

## Chimeric virions form during co-infection of V-073-resistant and V-073-susceptible viruses

We hypothesized that the lack of infectivity of virions made in V-073-treated cells was due to the formation of chimeric capsids (*Figure 1A*, bottom). To test explicitly whether capsid proteins encoded by V-073-susceptible and V-073-resistant genomes assemble into mixed capsid structures during co-infection (*Figure 4A*), which could be distinguished from separate, homogeneous structures (*Figure 4B*), we constructed viruses whose capsid proteins could be independently identified by antibodies. To this end, a FLAG epitope was engineered into the VP3-A24V virus and an HA epitope was engineered into wild-type, drug-susceptible virus (*Figure 4C*). Following a strategy employed in foot and mouth disease virus (*Seago et al., 2012*), the tags were inserted into a flexible, antibody-accessible loop in the assembled capsid structure. These constructs enabled immunoprecipitation of intact, viable virions that could be assessed for their drug susceptibility by soaking the virus in the capsid inhibitor before titering. In single-cycle infections, the FLAG-tagged virus retained its V-073-resistant phenotype, and the HA-tagged virus its drug susceptibility (*Figure 4D*). Anti-FLAG immunoprecipitation of lysates obtained from these single infections retrieved, as expected, FLAG-tagged drug-resistant but not HA-tagged drug-susceptible virus (*Figure 4E*) or proteins (*Figure 4F*, lanes 1 and 2, respectively). When the two lysates were pooled, only FLAG-tagged, V-073-resistant virus was recovered (*Figure 4E,F*, lane 3), demonstrating that the immunoprecipitation was specific for capsids harboring proteins from the V-073-resistant genome. When the drug-resistant, FLAG-tagged virus and the drug-susceptible HA-tagged virus were co-infected into cells and the anti-FLAG immunoprecipitation was performed, the immunoprecipitated virions displayed reduced infectivity when soaked in the capsid inhibitor prior to titering (*Figure 4E*). Furthermore, the anti-FLAG immunoprecipitate contained both FLAG- and HA-tagged VP1 proteins (*Figure 4F*, lane 4). Therefore, chimeric virions with reduced drug-resistance were formed in cells co-infected with V-073 susceptible and V-073-resistant viruses, even at the 1:1 ratio tested here. The formation of mixed capsids with reduced drug resistance is the most plausible explanation for the observed genetic dominance of V-073-susceptible variants and the blunted selection for drug-resistance during growth in mice.

## Design of a screen to identify dominant drug targets

For a viral product to be a dominant drug target, drug-susceptible virus must inhibit the growth of drug-resistant virus. Relying on this property and taking advantage of the intrinsic genetic diversity in 'wild-type' virus stocks, we used guanidine and V-073 to develop a simple MOI-based screen for

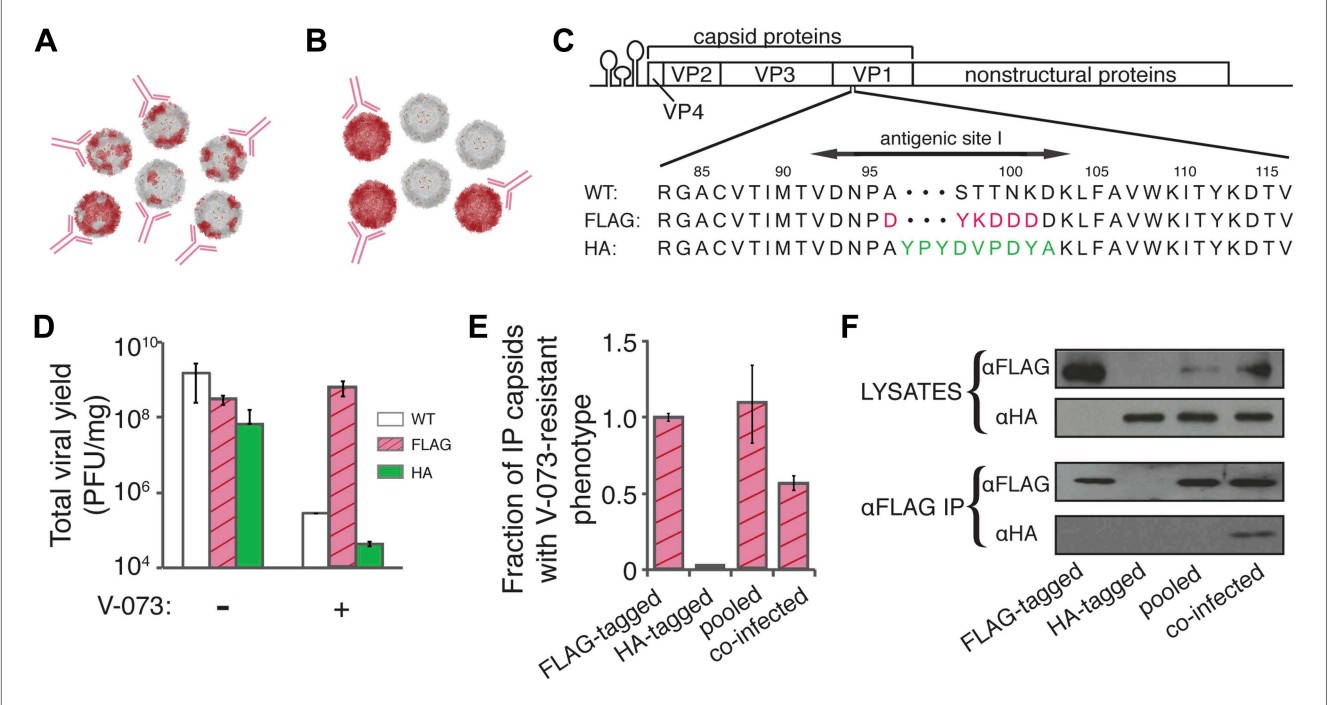

**Figure 4**. V-073-susceptible and V-073-resistant virus form mixed capsids during co-infection. (**A, B**) Depiction of 'chimeric' (**A**) and 'pooled' (**B**) viral preparations. Red subunits represent V-073-resistant, FLAG-tagged capsids, and grey subunits represent V-073-susceptible, HA-tagged capsids. Anti-FLAG antibodies are represented in pink. (**C**) Construction of FLAG- and HA-tagged virus. Sequence of FLAG and HA peptides inserted into antigenic site 1 in the poliovirus genome, replacing amino acids 96–101 and 97–102 of VP1, respectively. (**D**) Viral yield (PFU/ml) of the wild-type, FLAG-tagged and HA-tagged poliovirus variants in the absence or presence of V-073 over a single infectious cycle at an MOI of 10. Data represent means ± s.t.d. of two replicates. (**E**) Fraction of capsids with V-073-resistant phenotype in anti-FLAG immunoprecipitates as labeled. HA-tagged sample was below the limit of detection. Data represent means ± s.t.d. of two replicates. (**F**) Immunoblots of VP1 proteins collected by anti-FLAG immunoprecipitation of cell lysates prepared from infections indicated. 'Pooled' sample contains lysates from independently grown FLAG- and HA-tagged viruses, whereas 'co-infected' sample was obtained from cells infected with both FLAG- and HA-tagged variants. All variants were infected at an MOI of 10 PFU/cell.

compounds that inhibit dominant drug targets. Similar to the heterogeneity observed in virus stocks from mouse muscles, any stock of lab-passaged 'wild-type' virus contains pre-existing drug-resistant variants.

As depicted in *Figure 5A*, when a 'wild-type' virus stock infects cells at low MOI, any pre-existing drug-resistant variant in the stock will infect a cell in isolation (left panel) and have the opportunity to express its phenotype. Thus, it can be selected in the presence of drug, regardless of the drug target type. However, at high MOI, drug-resistant variants will occupy cells in concert with drug-susceptible variants (*Figure 5A*, middle). In cases of dominant drug targets, drug-susceptible viruses are genetically dominant and the inhibitor will prevent the outgrowth of drug-resistant variants. Consistent with this logic, when HeLa cells were infected at an MOI of 1 infectious virus per cell in the presence of either guanidine or V-073, the yields of guanidine-resistant and V-073-resistant viruses were similar (*Figure 5B*). At higher MOIs, the yield of guanidine-resistant progeny increased, as would be expected because more virus was present in the inoculum and nothing interfered with the outgrowth of the drug-resistant variants (*Figure 5C*, green). However, the yield of V-073-resistant progeny decreased at high MOIs (*Figure 5B* and *Figure 5C*, orange). To test whether the inhibition in growth of V-073-resistant virus was an intracellular event, as predicted if the formation of mixed capsids suppressed the drug-resistant phenotype, the virus stock was soaked in V-073 before the infections at different MOIs were performed (*Figure 5A*, right panel). When entry of the drug-susceptible viruses was thus inhibited, a dose-dependent increase in the yield of V-073-resistant virus was observed even at high MOIs (*Figure 5C*, dotted orange). The virus stock used was deliberately prepared from cells infected at low MOI to ensure that drug-resistant genomes were packaged in drug-resistant capsids. Thus, failure to

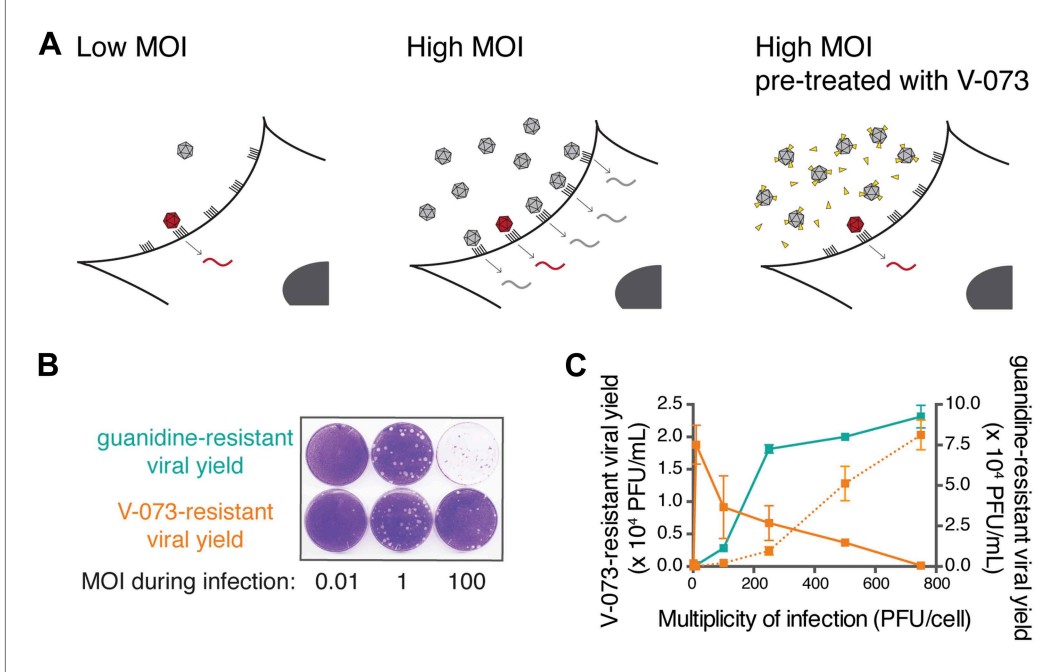

**Figure 5**. Assaying the dominance of drug-susceptible genomes via the MOI-dependent detection of drug-resistance. (**A**) Cartoon of proposed infection at low MOI (left) and high MOI without V-073 pretreatment (middle) or with V-073 pretreatment (right). Grey represents drug-susceptible variants, red represents drug-resistant variant and drug is depicted as yellow triangles. (**B**) Drug-resistant virus plaques following infections with wild-type Hispaniola strain at increasing MOIs in the presence of guanidine or V-073. (**C**) Drug-resistant viral yield (PFU/ml) in presence of guanidine (green line) or V-073 without pretreatment of viral stock (solid orange line) or with pretreatment (dotted orange line).

select V-073-resistant genomes at high MOIs could be overcome by preventing the entry of the drug-susceptible genomes.

This MOI-dependent emergence of drug resistance provides a rapid screen to identify inhibitors of dominant drug targets. Its utilization requires only the identification of a compound that inhibits viral growth, the ability to infect at MOIs above 1 virus/cell in cell culture, and the ability to titer drug-resistant viruses after infection at various MOIs. This assay does not require knowledge of the drug target, although such information will become even more interesting if it is determined that it is a dominant drug target.

## Discussion

In this study we have shown that a single inhibitor of a positive-strand RNA virus, capsid poison V-073, can both inhibit viral growth and suppress the emergence of newly arising drug-resistant variants during infection of mice and of cultured cells. Deliberate mixed infections performed in cell culture showed that the formation of chimeric capsids, which contained both drug-inhibitable and drug-resistant proteins, correlated with the suppression of drug resistance. These observations support the hypothesis that targeting the viral constituents of an oligomeric assemblage, such as a viral capsid, can lead to the genetic dominance of drug-susceptible genomes.

The magnitude of the suppression of V-073 resistance in murine infection was higher than expected given an assumption that all viruses in the inoculum contributed to the eventual infection. There are two potential sources of V-073-resistant viruses in infected mice. The first is the quaisispecies present in any inoculum. At an initial infectious dose of $10^7$ viruses per mouse, there should have been as many as $10^4$ pre-existing drug-resistant variants in the inoculum (*Holland et al., 1989*). Why were they unable to emerge from the infection? Given the effectiveness of the innate immune response, there is often a minimum effective dose of inoculated microorganism. For poliovirus in this mouse model, more than $10^4$ viruses are required to achieve any detectable viral growth in muscle tissue (data not

shown). This threshold predicts correspondingly fewer pre-existing drug-resistant variants in the effective inoculum. The second source of V-073-resistant viruses is the quasispecies addressed in the present study and depicted in *Figure 1A*: the newly arising variants formed within each individual cell by error-prone replication. Therefore, we conclude that V-073 was effective at suppressing resistance even as a monotherapy because V-073-susceptible variants suppress the growth of V-073-resistant variants as they arise intracellularly throughout the course of infection.

In the mouse experiments, the compounds were administered within hours of infection. Such rapid treatment is unlikely to occur in patients, except in cases where drugs are delivered prophylactically. However, a delay in treatment will not alter the degree to which dominant drug target inhibitors suppress the selection of newly arising resistant viral genomes from drug-susceptible parents. Late treatment may be directed at a population that contains 0.2% drug-resistant genomes, but many of these will have been trans-encapsidated by the other viral variants present in their cells of origin. Thus, for a capsid inhibitor, trans-encapsidation of newly arising drug-resistant genomes by the intracellular excess of drug-susceptible capsids should provide an additional mechanism to reduce the reservoir of drug-resistant genomes packaged in drug-resistant capsids.

The use of dominant drug targets is expected to reduce the number of antiviral compounds necessary for effective treatment of individuals and populations. In the case of V-073, the only poliovirus antiviral currently in human trials, there has been hesitation to use it as a monotherapy (*Collett et al., 2008*; *De Palma et al., 2008*). Our data suggest that V-073 may be sufficient for the effective treatment of poliovirus infection.

How can a dominant drug target be identified experimentally? The most direct way for a drug-susceptible virus to be genetically dominant over a drug-resistant one is the formation of mixed oligomers, as exemplified here. For poliovirus, co-transfection experiments identified known oligomers such as the viral capsid and the RNA-dependent RNA polymerase as viral products whose fitness could be suppressed by the presence of defective subunits (*Crowder and Kirkegaard, 2005*). Other candidates also emerged, however, whose mechanisms are less straightforward and remain under investigation: intramolecularly cleaving protease 2A and an RNA structure required for protein priming were also identified as potential dominant drug targets (*Crowder and Kirkegaard, 2005*). Here, the MOI-dependent of the outgrowth of drug-resistant variants in cell culture is suggested to be a rapid screen for whether an antiviral compound inhibits a dominant drug target or not.

How can dominant drug targets be identified a priori? All dominant drug targets should share a unique feature: binding of the drug to its target generates a product that is toxic to the drug-resistant variant. Three factors are required to accomplish this effect. 1) The drug-bound product must interact with proteins made by the drug-resistant genome; 2) the interaction between drug-bound products and resistant proteins must inhibit an essential step in the infectious cycle of the virus; and 3) the ratio of drug-susceptible to drug-resistant products synthesized in the cell must be high enough to poison the function of the resistant variant.

For viral structural proteins, criterion (1) is met through the direct binding of drug-inhibited and drug-resistant subunits within the capsid or core complex. However, a dominant drug target does not have to be an oligomeric protein itself. For example, poliovirus 2A protease has been illuminated as a potential dominant drug target, because when it fails to make an obligatorily intramolecular cleavage in the viral polyprotein, the uncleaved product is toxic (*Crowder and Kirkegaard, 2005*).

Criterion (1) dictates that the toxic product generated by drug binding retains the ability to interact with other proteins. This suggests one reason for a current lack of dominant drug target inhibitors. Small-molecule screens are often designed to detect loss of viral product function, rather than the deranged function required for the drug-bound, susceptible viral product to join and poison drug-resistant complexes. V-073, for example, hyper-stabilizes viral capsids. Thus, the drug-inhibited subunits can still insinuate themselves into capsid complexes. A theoretical capsid inhibitor that prevented assembly by binding at an intersubunit interface should display the characteristics of a non-dominant drug target inhibitor, because the drug-susceptible subunits would not have the opportunity to poison the drug-resistant subunits.

To fulfill criterion (2), we must identify stages of the viral infectious cycle that rely on the cooperation of proteins donated by multiple genomes. Capsid and cores are not the only such structures. Indeed, trans-assembly of highly oligomeric complexes is a hallmark of RNA viral infection, exemplified by the polymerase and polymerase-containing oligomers of positive-strand RNA viruses (*Crowder and Kirkegaard, 2005*; *Marcotte et al., 2007*; *Kemball et al., 2010*) and the

large nucleoprotein-RNA complexes that form during negative-strand RNA replication (Reviewed in reference *Morin et al. (2013)*).

For drug-susceptible subunits to inhibit chimeric complexes, they must be abundant enough to dictate the overall phenotype of the structure, fulfilling criterion (3). Again using the poliovirus capsid as an example, we can calculate the approximate number of susceptible subunits required to render the entire virion drug-susceptible. Assuming free mixing of drug-susceptible and drug-resistant capsids and relative abundances that reflect the ratio of their genomes, the data argue that about twenty subunits, or one third, of the capsid proteins must be drug-susceptible to poison the complex. For the abundance of drug-susceptible and drug-resistant proteins to mirror the representation of the RNA that encodes them, both genomes must both continue to amplify or must both be inhibited. This suggests another reason that inhibitors of structural proteins, which are predicted to be dominant drug targets, are often overlooked. V-073, for example, would not have been detected as a viable inhibitor in a screen that used a decrease in genome amplification, protein abundance or innate immune response activation during a single cycle of infection as a proxy for viral inhibition. Similarly, replicons, which cannot perform second rounds of infection, are useful tools for some kinds of screens but not for seeking inhibitors of many dominant drug targets.

The paradigm of targeting inhibitors to structures whose function can be inhibited by non-functional subunits has applications to other viruses, especially those with very high error rates due to the repeated amplification of RNA. Thus, capsids, cores and other oligomeric assemblages of rhinovirus, enterovirus 71, hepatitis C and dengue virus are likely to be dominant drug targets. This strategy may not apply to retroviruses, because the step with the least fidelity, reverse transcription, gives rise to a low copy number DNA intermediate. However, *bona fide* DNA viruses, present at a high copy number per cell, should also be amenable to this dominant drug targeting strategy. For other error-prone, intracellular genetic amplification events, such as the growth of bacterial and eukaryotic intracellular pathogens and the gene duplication associated with oncogenesis, the choice of oligomeric drug targets may also suppress the outgrowth of drug-resistant variants immediately after their inception. Deliberate targeting of oligomeric structures should reduce the numbers of inhibitors needed for effective treatment, but will require revision of decision-making processes during drug development.

## Materials and methods

### Virus strains and antiviral compounds

Poliovirus strains were either Mahoney type 1 poliovirus or the vaccine-derived type 1 Hispaniola strain (*Kew et al., 2002*). Solid guanidine hydrochloride from Sigma–Aldrich (St. Louis, MO) was dissolved to a concentration of 0.5 mM in water. Rupintrivir (trans-(4S,2R,5S,3S)-4-{2-4-(4-fluorobenzyl)-6-methyl-5'-[(5-methylisoxazole)-3-carbonylamino]-4-oxoheptanoylamino}-5-(2-oxopyrrolidin-3-yl)pent-2-enoic acid ethyl ester) from International Laboratory (San Bruno, CA) was dissolved to a concentration of 0.1 mM in DMSO. V-073 (1-[(2-chloro-4-methoxyphenoxy)methyl]-4-[(2,6-dichlorophenoxy)methyl]benzene) from *ViroDefense* (Rockville, MD) was dissolved to a concentration of 24 mM in DMSO.

### Infection of single variants, co-infections, MOI-dependent infections and titering

For all infections, HeLa cell monolayers were infected with poliovirus for 5 hr at 37°C, except for infections with the VP1-I194F Hispaniola variant which were carried out at 34°C to compensate for its slight temperature sensitivity. For growth of single variants in the presence and absence of drug, the MOI was 10 PFU/cell. For co-infections and the MOI-dependent selection experiments, the various MOIs are indicated on the figures and were performed in the presence of the drug of interest. Following the infectious cycle, intracellular virus was harvested by washing cells with PBS+ (PBS supplemented with 0.1 mg/ml $MgCl_2$ and $CaCl_2$), resuspending in 0.5 ml PBS+ and subjecting stocks to three freeze–thaw cycles before clearing the cell debris by low-speed centrifugation. Viral titer was determined by plaque assay as described previously (*Kirkegaard, 1990*) in the absence or presence of the drug as indicated. Drug concentrations were 0.5 mM guanidine, 0.1 µM rupintrivir and 24–47 µM V-073. For stocks incubated with V-073 prior to infection, the concentration was 0.47 µM. Co-infections were performed with the Mahoney strain except where indicated. The MOI-dependent selection was performed with the Hispaniola strain.

## Selection and cloning of drug-resistant variants

The guanidine-resistant variant, 2C-N179G has been previously published (*Pincus and Wimmer, 1986*). The rupintrivir-resistant variant was identified as follows: first, three rounds of plaque purification were performed with rupintrivir in the overlay. Independent plaque isolates were used to infect HeLa cell monolayers at MOIs of approximately 0.25 PFU/cell for 10 hr with rupintrivir in the medium. From this stock, a purified plaque was isolated and RNA was extracted and the sequence was determined. The 3C-G128Q mutation was identified and cloned into the T7pGempolio vector using the splicing by overlapping extension PCR method (*Lefebvre et al., 1995*), changing codon 128 in the 3C coding region from GGT to GAA. Viral RNA was made by linearizing 10 μg of the plasmid with EcoRI, phenol:chloroform extracting the DNA, and transcribing from the linearized DNA using the MEGAscript kit (Applied Biosystems, Foster City, CA). Viral RNA was transfected into HeLa cells with Lipofectamine 2000 (Invitrogen, Carlsbad, CA) using the manufacturer's protocol and agar overlays containing rupintrivir were added 4 hr post transfection to confirm the rupintrivir-resistant phenotype. Stocks of the virus were made in the presence of rupintrivir. V-073-resistant mutations in the Hispaniola strain have been published (*Kouiavskaia et al., 2011*; *Liu et al., 2012*). The V-073-resistant Mahoney type1 mutant was identified by three rounds of plaque purification with V-073 in the overlay. Viral RNA was extracted and reverse-transcribed as described above and the amino acid change was identified by sequencing with primer 5′-GTGGATTACCTCCTTGGAAATG-3′. Once the VP3-A24V mutation was identified it was introduced in isolation into the T7pGEM polio plasmid. Transfection, stock preparation, and confirmation of the compound-resistant phenotype were performed as described above for 3C-G128Q; the change in codon 24 of the VP3 coding region was from GCG to GTA.

## Cloning of FLAG and HA-tagged poliovirus

The FLAG tag was inserted into the VP3-A24V T7pGempolio vector using the splicing by overlapping extension (SOE) PCR method (*Lefebvre et al., 1995*). The forward primer was: 5′-ACCATTATGACC GTGGATAACCCAGACTACAAAGATGATGATGATAAGCTATTTGCAGTGTGGAAGA-3′ and the reverse primer was: 5′-TCTTCCACACTGCAAATAGCTTATCATCATCATCTTTGTAGTCTGGGTTATCCACGGT CATAATGGT-3′. The HA tag was inserted into the wild type T7pGempolio vector also by the SOE method using the forward primer: 5′-ATTATGACCGTGGATAACCCAGCTTATCCTTATGATGTCCCTG ATTATGCTAAGCTATTTGCAGTGTGGAAGATC-3′ and the reverse primer: 5′- ATCTTCCACACTG CAAATAGCTTAGCATAATCAGGGACATCATAAGGATAAGCTGGGTTATCCACGGTCATAAT-3′. RNA was in vitro transcribed, transfected and virus stocks were made from individual plaque isolates as described above.

## Preparation of samples for immunoprecipitation

HeLa cells were infected with either FLAG-tagged or HA-tagged virus at an MOI of 10 PFU/cell or they were co-infected with both viruses, each at an MOI of 10 PFU/cell. For the control 'pooled' sample, both viruses were grown separately and then pooled for the immunoprecipitation. Virus was grown for 7 hr at 37°C. Cells were washed once with TBS (50 mM Tris-Cl, 150 mM NaCl, pH 7.4) and then harvested in TBS. Stocks were made by freeze-thawing cells three times and cell debris was cleared by centrifuging at 2500 g for 5 min.

## Immunoprecipitation and analysis of tagged capsids

Capsids were immunoprecipitated using the Anti-FLAG M2 Affinity Gel (Sigma–Aldrich) according to the immunoprecipitation protocol. Beads were incubated with virus stock for 2.5 hr at room temperature followed by three 15 min washes with TBS before releasing the capsid from the antibody by competitive elution with the 3× FLAG peptide (Sigma Aldrich). Samples were prepared for SDS-PAGE gel by boiling for 5 min in protein sample buffer (final concentration: 62.5 mM Tris, 2% SDS, 10% glycerol, 100 mM DTT, 0.0005% bromophenol blue) and analyzed on a 12% SDS-PAGE gel. Proteins were visualized by immunoblot. For the anti-FLAG immunoblot, a monoclonal anti-FLAG M2 antibody was used (Sigma–Aldrich) at a 1:1000 dilution in 2% BSA in TBST (TBS with 0.05% Tween). For anti-HA immunoblot, the THE HA Tag Antibody (GenScript, Piscataway, NJ) was used according to the manufacturer's protocol.

## Mouse infections and viral analysis

Previously described transgenic cPVR mice (*Crotty et al., 2002*) and *Tnfr1*−/− *PVR*+/+ mice (described below) were infected with Mahoney type 1 virus by intramuscular inoculation. Mice were inoculated with 1 × 10⁶ PFU for all experiments except the extended drug-treatments in which *Tnfr1*−/− *PVR*+/+

mice were inoculated with $1 \times 10^7$ PFU. For guanidine treatments, female mice were injected subcutaneously with 0.5 ml of 100 mM guanidine in PBS twice daily. Control mice were injected with PBS only. V-073 was delivered to mice by oral gavage in 100 μl corn oil. Control mice were given DMSO in corn oil. To titer virus in muscle samples taken, calf muscles were harvested, homogenized in 1 ml PBS$^+$, subjected to one round of freeze-thaw, cleared by low-speed centrifugation and titered as described above. V-073-resistant virus was titered by infecting between three and twenty plates at an MOI of 0.02 PFU/cell and adding 24 μM V-073 in the overlay; low amounts of virus were used to minimize toxicity from wild-type virus. As many plates as needed were infected in order to observe at least ten V-073-resistant plaques. If fewer than ten plaques were observed, the observed value was assumed to be $k$ and the Poisson distribution was used to determine the highest possible λ value with a 5% likelihood (*Poisson, 1837*). *Tnfr1$^{-/-}$* mice that express the human poliovirus receptor were derived by crossing cPVR mice with *Tnfr1$^{tm1Imx}$*/J mice (JAXS, Bar Harbor, ME) and breeding both the transgenic PVR and the *Tnfr1$^{-/-}$* allele to homozygosity.

## Fitness of drug-resistant variants in mice

Stocks of mouse-adapted wild-type and drug-resistant virus were made as follows: Using viral stocks obtained from mouse muscle tissues harvested 4 days post infection, plaque assays were performed by adding overlays containing V-073, guanidine or no drug. Eight to ten large plaques were picked for each condition, pooled and amplified for two cycles in the presence of V-073 or guanidine or in the absence of drug, depending on the conditions under which the plaques were originally grown. *Tnfr1$^{-/-}$ PVR$^{+/+}$* mice were infected with $1 \times 10^6$ PFU of these mouse-adapted wild-type, V-073-resistant or guanidine-resistant virus stocks. 4 days post infection, muscles were harvested and made into viral stocks for titering (as outlined above).

## Statistical analysis

For co-infection experiments, a two-tailed equal variance Student's $t$ test was used to determine p values. For co-infections, n = 3 for all viral yield data and each point was compared to the yield of drug-resistant virus in the absence of drug-susceptible virus. For total viral yield in mouse muscles, a Mann–Whitney $t$ test was used. For V-073-treated *Tnfr1$^{-/-}$* samples, all days were combined for analysis by converting values to a fraction of the control mean (i.e. titer of sample/control average). For the frequency of drug resistance, a two-tailed equal variance Student's $t$ test was used to determine p values.

## Acknowledgements

We thank Roberto Mateo, Ryan Ritterson and Peter Sarnow for many helpful discussions and critical reading of the manuscript, and Julie Theriot for advice on statistical analysis. This work was supported by the World Health Organization, an NIH training grant (EJT) and the NIH Director's Pioneer Award (to KK).

## Additional information

### Competing interests

MSC: President of ViroDefense, Inc., the sponsoring firm for development of V-073. The other authors declare that no competing interests exist.

### Funding

| Funder | Grant reference number | Author |
| --- | --- | --- |
| National Institutes of Health | Pioneer Award | Karla Kirkegaard |
| World Health Organization | Global Poliovirus Eradication Initiative Grant | Karla Kirkegaard |
| National Institutes of Health | Training Grant | Elizabeth J Tanner |

The funders had no role in study design, data collection and interpretation, or the decision to submit the work for publication.

### Author contributions

EJT, KK, Conception and design, Acquisition of data, Analysis and interpretation of data, Drafting or revising the article; H-L, Analysis and interpretation of data, Contributed unpublished essential data

or reagents; MSO, MP, Drafting or revising the article, Contributed unpublished essential data or reagents; MSC, Acquisition of data, Drafting or revising the article

### Author ORCIDs

Elizabeth J Tanner, http://orcid.org/0000-0001-9445-6620

### Ethics

Animal experimentation: Mice used in these studies were bred and housed under specific pathogen-free conditions at the Stanford University animal care facility, which is accredited by the Association for the Assessment and Accreditation of Laboratory Animal Care, Int. All experiments were approved by Stanford's Institutional Animal Care and Use Committee (Administrative Panel of Laboratory Animal Care). The Assurance number for this panel is A3213-01; the Protocol ID is 9296. For survival studies, mice were euthanized when moribund or upon initial signs of paresis/paralysis.

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
