## [Decision Letter]

Thank you for sending your work entitled “Dominant Drug Targets Suppress the Emergence of Antiviral Resistance” for consideration at *eLife.* Your article has been favorably evaluated by Richard Losick (Senior editor), a Reviewing editor, and 2 reviewers.

The Reviewing editor and the reviewers discussed their comments before we reached this decision, and the Reviewing editor has assembled the following comments to help you prepare a revised submission.

In this manuscript, Tanner et al. show that targeting poliovirus capsid with a compound (V-073) effectively suppressed the emergence of drug-resistant variants in cell cultures and, more notably, during the infection of murine models of poliovirus. By contrast, targeting viral proteins which do not form higher order oligomers, such as the 3C protease or the 2C RNA-dependent NTPase of poliovirus, failed to inhibit newly arising drug-resistant virus. The authors demonstrated that formation of chimeric virions containing a mixture of drug-susceptible and drug-resistant capsids was responsible for the observed genetic dominance of drug-susceptible viruses in the studies, and they propose targeting such a 'dominant drug target' would suppress the outgrowth of naturally occurring drug-resistant variants in general. They also examined how the dose of inoculating virus influences the isolation of drug resistant variants in cell culture; which might form a basis for a general methodology to identify dominant drug targets.

The study tackles an important aspect of antiviral drug therapies: the emergence of resistance. Overall, the conclusions of the study are well supported by the data presented and the concept itself has significant potential to influence the development of inhibitors as well as the use of monotherapy in patients.

1) Figure 3 represents one of the most critical pieces of data of the entire paper. The data are clearly presented and show that the wild type dominantly interferes with the V-073 resistant virus. The authors contend that because of the oligomeric nature of the capsid that in any given cell a drug sensitive capsid will dominantly interfere with a drug resistant capsid. This raises the question of how the authors were able to select a resistant mutant? There are two answers presented in the manuscript, one is that due to the error prone nature of RNA replication natural variants will be generated in the absence of drug that turn out to be drug resistant. A second answer is provided in Figure 5 where the authors study the impact of MOI on isolation of resistant variants. It might help some readers to provide a more explicit explanation of this.

2) The manuscript would also benefit from discussion of the proposed mechanism by which the WIN compounds block infection. How many copies of drug sensitive or resistant capsid protomers are required to render the assembled virus sensitive or resistant? Can the authors discuss/speculate particularly in relation to how this influences this region of capsid as a dominant drug target?

3) A more thorough description of the lack of additional drug candidates that could be used for studying dominant drug target may help strengthen authors' conclusions.

4) In the section of Mouse infections and viral analysis, please clarify at what time point the drugs were applied, can the authors speculate how this experimental scheme would be relevant to use V-073 in human patients?

---

## [Author Response]

*1)*
Figure 3
*represents one of the most critical pieces of data of the entire paper. The data are clearly presented and show that the wild type dominantly interferes with the V-073 resistant virus. The authors contend that because of the oligomeric nature of the capsid that in any given cell a drug sensitive capsid will dominantly interfere with a drug resistant capsid. This raises the question of how the authors were able to select a resistant mutant? There are two answers presented in the manuscript, one is that due to the error prone nature of RNA replication natural variants will be generated in the absence of drug that turn out to be drug resistant. A second answer is provided in*
Figure 5
*where the authors study the impact of MOI on isolation of resistant variants. It might help some readers to provide a more explicit explanation of this*.

Yes, the relevant variation pre-exists, and the drug-resistant viruses can be selected if they infect a cell singly, at low MOI. This has been explained earlier in the paper.

2) The manuscript would also benefit from discussion of the proposed mechanism by which the WIN compounds block infection. How many copies of drug sensitive or resistant capsid protomers are required to render the assembled virus sensitive or resistant? Can the authors discuss/speculate particularly in relation to how this influences this region of capsid as a dominant drug target?

Encouraged by the reviewers, we calculated the number of drug-susceptible subunits needed to render a chimeric virion drug-susceptible. The number (approximately 20 out of 60) and the assumptions (free mixing in the cell and equivalent synthesis per genome) are discussed in the manuscript.

*3) A more thorough description of the lack of additional drug candidates that could be used for studying dominant drug target may help strengthen authors' conclusions*.

These thoughts are now elaborated in the revised manuscript.

4) In the section of Mouse infections and viral analysis, please clarify at what time point the drugs were applied, can the authors speculate how this experimental scheme would be relevant to use V-073 in human patients?

The delivery of compounds immediately after infection is now described. A discussion of the relevance of this to human infections is also included.